# Recent Advances in Electro-Optic Response of Polymer-Stabilized Cholesteric Liquid Crystals

**DOI:** 10.3390/ma16062248

**Published:** 2023-03-10

**Authors:** Kyung Min Lee, Zachary M. Marsh, Ecklin P. Crenshaw, Urice N. Tohgha, Cedric P. Ambulo, Steven M. Wolf, Kyle J. Carothers, Hannah N. Limburg, Michael E. McConney, Nicholas P. Godman

**Affiliations:** 1Air Force Research Laboratory, Materials and Manufacturing Directorate, Wright-Patterson Air Force Base, Dayton, OH 45433, USA; 2Azimuth Corporation, Beavercreek, OH 45431, USA; 3Department of Materials Science and Engineering, Texas A&M University, College Station, TX 77843, USA

**Keywords:** cholesteric liquid crystals, electro-optic response, polymer stabilization, ion-trapping mechanism

## Abstract

Cholesteric liquid crystals (CLC) are molecules that can self-assemble into helicoidal superstructures exhibiting circularly polarized reflection. The facile self-assembly and resulting optical properties makes CLCs a promising technology for an array of industrial applications, including reflective displays, tunable mirror-less lasers, optical storage, tunable color filters, and smart windows. The helicoidal structure of CLC can be stabilized via in situ photopolymerization of liquid crystal monomers in a CLC mixture, resulting in polymer-stabilized CLCs (PSCLCs). PSCLCs exhibit a dynamic optical response that can be induced by external stimuli, including electric fields, heat, and light. In this review, we discuss the electro-optic response and potential mechanism of PSCLCs reported over the past decade. Multiple electro-optic responses in PSCLCs with negative or positive dielectric anisotropy have been identified, including bandwidth broadening, red and blue tuning, and switching the reflection notch when an electric field is applied. The reconfigurable optical response of PSCLCs with positive dielectric anisotropy is also discussed. That is, red tuning (or broadening) by applying a DC field and switching by applying an AC field were both observed for the first time in a PSCLC sample. Finally, we discuss the potential mechanism for the dynamic response in PSCLCs.

## 1. Introduction

Calamitic liquid crystals (LCs) are rod-shaped organic molecules that exist between amorphous liquids and solid crystals. These anisotropic rods spontaneously self-assemble into a variety of periodic structures known as mesogenic phases. LCs are, arguably, one of the most well-studied systems for manipulating light, as their unique optical properties arise from a positional order that endows these materials with anisotropy. There are several types of LC phases and they can be distinguished by their textures and optical properties. Nematic LCs in particular display 1D long-range order, wherein the mesogens align along their long axis, but each individual compound is free flowing and randomly distributed throughout the bulk. By adding small amounts of chiral molecules into the nematic LCs, a chirality transfer to the mesogens is induced, creating a chiral nematic or cholesteric phase. The anisotropic nature and inherent dielectric anisotropy of LC materials enables the natural ordering to be manipulated using external stimuli. The electro-optic behavior of LCs has been utilized for display technology since the discovery of the first LC display in the 1960s [1]. Since then, LCs have been utilized in other technologies including visors, smart windows, medical devices, and thermometers.

Cholesteric liquid crystals (CLC) are 1D photonic crystals that exhibit a circular rotation of the LC director along a helical axis. The chiral nature of the self-assembled CLC is determined by the handedness of the chiral dopant, thus the helical rotation of a CLC is either right-handed or left-handed depending on the optical activity of the chiral dopant [2,3]. When unpolarized light is exposed to a CLC, a maximum of 50% circularly polarized light (CPL) of the same handedness as the CLC is reflected and 50% CPL of the opposite handedness is transmitted, which is called selective reflection (Figure 1). The position and bandwidth of the selective reflection are related to the pitch ($P_{0}$), and are expressed as $\lambda_{0}=\bar{n}\cdot P_{0}$ and $\Delta \lambda =\Delta n\cdot P_{0}$, where $\lambda_{0}$ is the center of the wavelength, Δλ is bandwidth, $P_{0}$ is the natural cholesteric pitch length, $\bar{n}=\left(n_{e}+2n_{0}\right)/3$ is the average refractive index, and $\Delta n=n_{e}-n_{0}$ is the birefringence of the LC. It should be noted that $n_{e}$ and $n_{o}$ are the extraordinary and ordinary refractive indexes of the LC, respectively. Since the CLC is formed by mixing a chiral dopant with a nematic LC, the position of the reflection band can be tuned by the chiral dopant concentration added to the LC mixture. The $P_{0}$ of the CLC is the distance along the helical axis that the director changes by 2π and is inversely proportional to the concentration of the chiral dopant. The reflection bandwidth of CLCs in the visible wavelength region is tens of nanometers (50–100 nm) and broadens at longer wavelengths; e.g., near infrared and beyond.

LCs are classified by their dielectric anisotropy (Δε), where $\Delta \varepsilon =\varepsilon_{\left|\right|}-\varepsilon_{⊥}$. This is representative of the difference between the permittivity parallel to the long axis ($\varepsilon_{\left|\right|}$) and the permittivity of the short axis ($\varepsilon_{⊥}$) of the liquid crystal. LCs with a positive Δε will orient parallel to the direction of an electric field, while LCs with a negative Δε align perpendicular to the electric field direction [4]. The properties of a CLC system are highly dependent on the Δε of the chosen LCs. An example is a planar-aligned CLC system, which can be prepared to have an initial state of either reflective or scattering [5,6,7,8,9,10,11,12,13,14,15,16,17,18,19,20,21]. A CLC with positive Δε LCs can be switched to an optically transparent state by applying an electric field, which orients the liquid crystal molecules in the homeotropic orientation. After the field is removed, the CLCs in the homeotropic state relax back to the initial planar cholesteric state through the metastable focal conic state [5,6,7]. Since this relaxation process is very slow, polymer stabilization methods are used to improve the relaxation kinetics through strong anchoring between the polymer network and the CLC medium [8,9,10]. In contrast, CLCs with a negative ∆ε do not experience rotation in the planar state when applying an electric field.

Another advantage of LCs is the ability to stabilize the molecular orientation through polymer stabilization. Polymer-stabilized liquid crystals (PSLCs) consist of liquid crystals and a polymer network that is formed via in situ polymerization of reactive monomers dispersed throughout the LC system [22,23,24]. When the LCs are in the cholesteric phase, the LC polymer-composites are called polymer-stabilized cholesteric liquid crystals (PSCLCs). In both PSLCs and PSCLCs, small amounts of polymer are used to form the stabilizing network, typically below 10 wt.%. The polymerization is performed in an ordered phase of the mixture, preserving the initial alignment of the reactive mesogens in the resulting polymer network. Therefore, the polymer chains can act as alignment surfaces distributed throughout the thickness direction to stabilize the LC. This is evident when the temperature increases near the clearing temperature of the LC in PSLCs; at these temperatures, the order parameter sharply decreases [25,26], but complete isotropization is not achieved. Order is maintained due to the LC molecules strong anchoring to the surface of the polymer network. Therefore, these LC-polymer composites exhibit temperatures and electro-optic responses that differ from the bulk LC phase [22,24,27,28]. For example, in PSLCs and PSCLCs with positive dielectric anisotropy, the LC molecules can be reoriented by applying an electric field, the threshold voltage of which is higher than that of bulk CLC [3,5,25,27,29,30]. Due to the restoring force exerted by the polymer network, recovery to the initial state is much faster in the polymer-stabilized CLCs than in the bulk system.

Polymer networks can be used as three-dimensional scaffolds for aligning low molecular weight LC molecules. In this process, the LC molecules can be removed from the PSCLC composite using organic solvents, leaving structurally chiral insoluble polymer fractions. Then, the system can be refilled with other liquid crystal molecules [3,22,31]. These ordered polymer networks can strongly align non-mesogenic molecules or overcome the initial handedness of chiral LCs. When the polymer scaffold is refilled with CLC of the opposite handedness, the refilled PSCLC has the handedness of the polymer template, indicating that the refilled CLC fluid follows the handedness of the polymer template [32]. This washout/refill method can be used to prepare CLC-polymer composite systems with unique properties that are not observed in natural systems, such as hyper-reflective devices that reflect both right-handed and left-handed circularly polarized lights in a single cell [31,33,34,35].

The polymer network of PSCLCs can also lock the pitch of a CLC, and it affects various optical responses. Stabilizing low molar mass CLCs with positive Δε using polymer networks allows for fast and bistable switching between reflective and scattering textures [6,7,8,9,10,11,12,13,14,15,16,17,36,37,38]. These composite systems are described as PSCLCs, but, in the past, these systems were also known as polymer-stabilized cholesteric textures (PSCTs). PSCTs have two types of switching modes: normal mode and reverse mode. In normal mode, the switching from scattering to clear or scattering to reflective states occurs [5,18,19], and in reverse mode, the switching from reflective to scattering state is observed [5,20,21]. The normal or reverse operating mode is mainly determined by the sample preparation conditions. Applying an AC electric field during photopolymerization to prepare the normal mode samples aligns the LCs in the direction perpendicular to the cell. When the applied electric field is turned off after polymerization, the PSCLC forms the hazy focal conic texture. Therefore, the initial state of this sample, the off-state, has a scattering state. Upon applying an electric field, the positive Δε LCs reorient to the homeotropic state, also referred to as the clear state. To prepare the reverse mode sample, the mixture is polymerized in the off-state to obtain a sample with a stable reflective state in the off-state. However, when the reflection band is outside the visible range, the sample is optically transparent. Intermediate controls of scatter, or various gray scales, have also been reported [6,7,11,12].

In contrast, negative Δε LCs in a planar state do not respond to an applied electric field and have no switching response. Over the past decade, the dynamic response of PSCLCs with negative Δε has been extensively studied by the Air Force Research Laboratory (AFRL) liquid crystal team, including bandwidth broadening [39,40,41,42,43], red tuning [44,45,46,47], blue tuning [48], and switching [49,50]. The proposed mechanism for the electro-optic responses is based on the ion-mediated deformation of the polymer network that stabilized the CLC medium. The application of a DC field induces the migration of ions trapped in the polymer network, leading to the deformation of the polymer network toward the negative electrode. The LC host anchored on the deformed polymer network is also deformed, resulting in a cholesteric pitch variation across the cell [39,41,46]. Recently, a reconfigurable EO response of a single positive Δε PSCLC sample, such as switching and red-shifting tuning or switching and bandwidth broadening responses, by properly adjusting the AC and DC fields has been reported [51]. The redshift of the reflection band of the positive Δε PSCLC is due to strong anchoring of the LCs on the polymer network. This review summarizes the current understanding and potential mechanisms for the dynamic response of PSCLCs studied by the AFRL liquid crystal research team over the past decade.

## 2. Electro-Optic Response in PSCLCs

### 2.1. Bandwidth Broadening

Bandwidth broadening in PSCLCs has been an intensive research topic over the last decade from both an academic and practical application perspective. While a number of approaches exist to induce bandwidth broadening, electrically induced responses in PSCLCs are of great interest [42,52,53,54,55,56,57,58,59,60]. PSCLCs employed for such studies are typically comprised of CLCs with negative Δε and an anisotropic polymer network with a strong alignment effect on the LC [58,61]. The polymer network entraps ions during photopolymerization, and the movement of these ions via an applied electric field leads to deformation of the polymer network. The deformation leads to a change in pitch for the anchored LCs. The bandwidth broadening phenomenon is generally attributed to the non-uniform changes in these pitches.

In 2011, Bunning and his team first reported the electrically induced bandwidth broadening response of PSCLCs. Electro-optic studies have highlighted the effect of trapped charges on the polymer network of PSCLCs [39]. The distortion of the polymer network upon the application of the electric field led to the variation in pitch length and subsequent bandwidth broadening. Detailed studies by Yang and coworkers [58] showed that polymer networks and ion density were most influential on bandwidth broadening in PSCLCs. The authors also reported that the choice of photoinitiator had a large effect on the generated ion density, but not the polymer network, as evidenced by Scanning Electron Microscopy studies. The chiral dopant and monomer functionality were found to influence the polymer network as well. An optimal choice of the aforementioned parameters is thus needed for the desired bandwidth broadening in PSCLCs.

Duan et al. reported on a step-wise photopolymerization process utilizing two kinds of UV light, specifically 254 nm and 365 nm, as well as cationic and free radical photo-initiators [62]. The intensity of the light source and irradiation time were used to control bandwidth broadening in PSCLCs. In one case, broadening was observed from 916 nm to 1460 nm due to an increase in 254 nm UV light intensity from 0.5 mW cm^−2^ to 3.5 mW cm^−2^.

A study by Nemati et al. found a correlation between the electrical resistivity and reflection bandwidth broadening of a PSCLC [61]. Higher ion densities led to a low resistivity in the cell, resulting in bandwidth broadening. The authors subsequently explored the impact that alignment layer thickness has on impedance in the cell and experimented with different additives to increase conductivity. They found that a thinner alignment layer led to bandwidth broadening, while increasing the thickness of the alignment layer was detrimental to bandwidth broadening. This was attributed to the polyimide acting as an insulating layer, which increased the impedance of the cell. The authors also reported that additives with a phenyl functional group were found to enhance broadening, while an additive containing two carboxylic acid functional groups suppressed broadening. The incorporation of esters into the reactive mesogen monomer may also improve the broadening performance, as it has previously been reported that the ester moieties of RM257 act as effective ion trapping groups [63].

Khandelwal et al. demonstrated electrically induced large-magnitude bandwidth broadening using a long and flexible ethylene glycol crosslinker twin molecule. This is due to the molecule enhancing the ability of the polymer-stabilizing networks to trap cationic impurities, and these ion-trapped PSCLCs show a nine-fold enhanced reflection bandwidth [56].

Lee et al. reported color-tunable mirrors based on reflection bandwidth broadening in PSCLCs [42]. The authors prepared a PSCLC device that switches from being selectively reflective to forming a broadband reflection mirror upon the application of an 80 V DC field. This is visually represented in Figure 1a, where, in the off-state (left), the word “Mirror” is visible behind the PSCLC device. When the device is turned on (right), “Mirror” becomes obscured and a reflection of the image is observed. It is important to note that the reversible behavior of the PSCLC takes several seconds to occur. The authors also carried out the effect of viscoelastic behavior of a polymer structure prepared from LC monomers with various alkyl chain lengths as a function of the electric field. Bandwidth broadening from 100 nm to nearly 600 nm, symmetric about the center of the reflection notch, was realized. The transmission spectra in Figure 1b show the broadening effect upon application of a DC field in the range of 0–140 V. The length of the flexible methylene spacers in the LC monomer unit (n = 3, 6, and 11) also influences the magnitude of the voltage required to achieve bandwidth broadening. Interestingly, the threshold voltage for bandwidth broadening was lower for LC monomers with longer spacers, and this was attributed to the viscoelastic properties of the polymer network. The effect of alkyl chain length on the magnitude of bandwidth broadening is summarized in Figure 1b(iv).

The reports on reflection bandwidth broadening in PSCLCs support the hypothesis that the behavior is controlled by the polymer network. The polymer network both anchors the low molecular weight LCs and entraps ions. When a strong field is applied to a PSCLC, the movement of trapped ions leads to the deformation of the polymer network, which in turn expands and contracts the pitch of anchored CLCs. The availability of ions in the system stems from the choice of initiator, additives, and curing/photopolymerization conditions. Recent efforts in this area have focused on the development of switchable glasses, which may have applications for smart window technology [64,65,66,67].

### 2.2. Reflection Notch Tuning

Another dynamic mechanism of PSCLCs is responsive tuning of the selective reflection color through the application of external stimuli. Electrical, mechanical, optical, and thermal inputs have been used to tune the selective reflection peak. Generally, the use of a small electrical field to induce color change is preferred. Nematic LCs with a negative Δε are utilized for tuning behavior in PSCLCs, as the application of a DC field will not reorient the LC mesogens. This enables tuning of the reflection notch instead of switching behavior.

Previous methods used to electrically induce color changes have employed cell architectures containing interdigitated electrodes [68]. Interdigitated electrodes lead to non-uniform coloration across the cell due to the complex geometry of the electrodes [69,70]. In these systems, focal conic domains, which readily scatter light, form. However, polymer stabilization of these systems has largely corrected this issue regarding the complex geometry of the electrodes and the scattering of the focal conic texture [40,44,54,60]. Another downside to interdigitated electrodes is the need for large electric fields to induce the color tuning of the CLC.

The complexity of cell architectures containing interdigitated electrodes paired with large electric fields has driven the use of conventional cell architectures. Examples of small-scale color tuning using conventional cells include annealing defects in a CLC, which enabled the switching of a laser on and off through the application of a DC field [71]. Color tuning was also achieved through tilting of LCs with positive Δε along their helical axis [72]. Piezoelectric compression of a cell containing CLCs led to a compression of the pitch, which resulted in color tuning [73,74]. Electrostatic deformation of a CLC also enabled pitch compression, resulting in blue tuning of the reflection notch [75]. These approaches lead to small-scale color tuning of the selective reflection of a CLC. In order to achieve a larger tuning range polymer stabilization must be used.

Recently, tunable notch and bandpass filters capable of spanning the entire visible spectrum with simple device architectures have been reported by using the oblique helicoidal cholesteric phase, also referred to as the heliconical cholesteric phase [68]. The heliconical cholesteric phase was attained by mixing the dimeric mesogens CB7CB and CB11CB, nematic liquid crystal 5CB, and left-handed chiral dopant S811. The unique structure of the heliconical cholesteric state endows the material with dielectric torque upon the application of an electric field, the periodicity of the system changes, resulting in tuning of the selective reflection. Tuning of the reflection notch from 465 to 637 nm was observed at low voltages of 1.6–2.1 V μm^−1^, respectively. Similarly, electrically tunable optical bandpass filters were fabricated. The efficiency of the bandpass filters was 15% due to reflection losses from the cholesteric and attenuation losses of the polarizer.

Though tuning of the selective reflection is attainable using the electro-optic response of bulk CLCs, large-range reflection tuning often requires polymer stabilization. The implementation of polymer stabilization has enabled large magnitude color tuning, greater than 300 nm, while utilizing low to moderate electric fields. In 2005, Yu et al. from Kent Optronics reported a PSCLC system that exhibited large-magnitude reflection notch tuning of 300 nm [76]. A cross-linkable monomer, photoinitiator, and low molecular weight LC with a positive Δε were employed to generate a highly cross-linked system. Upon the application of an electric field, the polymer network remained unresponsive but provided stabilization, enabling the rotation of the LCs molecules. Rotation of the LCs induced effective refractive index changes, and thus, tuning of the reflection notch. Interestingly, the PSCLC could also be doped with a laser dye, PM-597, to generate a CLC laser with a tunable lasing wavelength. The system exhibited a 33 nm blue shift in the lasing wavelength over a voltage range of 324 V [76].

Large-magnitude color tuning can also be achieved through delocalization of the polymer-stabilizing network to induce pitch modulation. In these systems, the structural chirality of the polymer network is manipulated by applying a DC field through an electrophoretic mechanism. In order to induce deformation of the polymer network, a negative Δε LC is required. The negative Δε LCs ensure that the molecular reorientation does not occur. This enables the movement of the polymer network, and therefore, the large magnitude color tuning. This phenomena was first reported by McConney et al. in a PSCLC containing a low molecular weight negative Δε LC that exhibited red shifting behavior [44]. By applying a DC field of 8 V µm^−1^, the reflection notch was tuned to nearly 300 nm. This corresponded to a change in the pitch length of 55% the original pitch length [45]. The PSCLC had facile and reproducible color tuning throughout the visible region into the near IR. However, high polymer concentrations up to 20 wt.% were required. The high polymer content resulted in poor transmission values.

To improve the low transmission values of the PSCLC, Lee et al. reduced the overall polymer content to 6–8 wt.%, resulting in increased transmission values [46]. In this system, the reflection notch was tuned from 700 to nearly 2500 nm upon application of 160 V of a DC field. The PSCLC device maintained high out-of-band transmission values across the entire tuning range, as shown in Figure 2a. Tuning throughout the visible spectra was also achieved in the PSCLC by increasing the chiral dopant concentration. However, appreciable creep in the reflection notch position was observed due to the lower polymer content. At lower polymer concentrations, around 5 wt.%, the creep was more pronounced, while samples with 8 wt.% polymer exhibited higher stability. Figure 2b demonstrates the ability of the PSCLC to adjust the notch position of the selective reflections. The electric field was directly applied to 50 V DC and then maintained for 30 min. When the field was turned off, the reflection band was relaxed back to its original position. The same procedure was performed for 200 nm, 300 nm, and 400 nm displacements of the reflection notch. The inset of Figure 2b shows the POM images of the reflection color as the DC voltage was increased from 0 V to 75 V. During the 30 min that the electric field was maintained, the reflection notch continued to move (creep) and the magnitude of the creep depended on the strength of the electric field. However, this is mainly because the degree of deformation of the polymer network is proportional to the length of time of the applied DC field.

PSCLCs with tunable reflection notch behavior have been utilized to generate a dynamic bandpass filter that operates in the mid-wave infrared region (MWIR). Here, Worth et al. fabricated a PSCLC containing a 6.5 wt.% diacrylate monomer RM82 and a negative Δε LC MLC 2079 that exhibited large magnitude tuning across the MWIR [77]. In order to achieve a saturated reflection in the MWIR, a cell thickness of 50 μm was required. Despite the larger cell gap, the PSCLC retained out-of-band transmission values near 70% due to the low polymer content. The PSCLC filter was able to tune over 2 μm from 2.5–4.9 μm by applying a DC field of 110 V. By adjusting the photo-polymerization procedure of the same PSCLC formulation, a bandwidth broadening MWIR filter was able to be generated. Therefore, a dynamic PSCLC exhibiting tunable and broadening behavior in the MWIR was produced by manipulating the structural chirality of the PSCLC polymer network.

Electromechanical distortion of the polymer stabilizing network has largely been utilized for the red shift of the reflection notch upon the application of a DC field. In 2017, the blue shifting of the reflection notch was realized by Lee and coworkers [48]. A PSCLC with a reflection notch centered at 700 nm was blue-shifted to 430 nm by applying a 35 V DC field, the transmission spectra for this sample is shown in Figure 3a. Photographs of the reflection colors in the PSCLC alongside wavelength positions are shown in Figure 3b. The blue shifting behavior was determined to be highly sensitive to the photo-polymerization conditions and the photoinitiator. When the PSCLC was cured at lower intensities of 250 mW cm^−2^ of 365 nm UV irradiation, red shifting was observed after three minutes, and bandwidth broadening was observed between 5–10 min, respectively. Only after the sample was cured for 15 min was blue shifting behavior realized. If the sample was cured at higher intensities of 700 mW cm^−2^ of 365 nm UV irradiation, blue shifting was observed after 10 min of curing. The photoinitiator was also crucial to achieving blue shifting of the reflection notch. The authors determined that morpholino-keto-type initiators, such as Irgacure 369 and 907, were able to induce blue shifting behavior with concentrations as low as 0.5 wt.% after 30 min of irradiation at 250 mW cm^−2^ of 365 nm UV light. When a non-morpholino-keto-type initiator was used, specifically, Irgacure 651, red shifting behavior was observed even at concentrations ranging from 0.5–5 wt.% under similar curing conditions.

Utilizing the dynamic behavior enabled by electromechanically distorting the polymer network of PSCLCs, Lee et al. produced a PSCLC with complete optical reconfigurability [51]. Figure 4a shows the DC-field-induced large-scale red tuning of the reflection band of a CLC composed of 6 wt.% SL04151 and the positive Δε LC E7. At appropriate AC field strengths, the orientation of the LC to the polymer network can be overcome, resulting in a switching of the positive Δε LCs and loss of the reflection notch (Figure 4b). Reversible and repeatable switching was achieved by applying a 150 V (1 kHz) AC field. A DC field was also concurrently applied to the system, yielding an optical element with a tunable and switchable reflection notch. This switchable and tunable behavior is shown in Figure 4c. The high anchoring strength of the LC to the polymer network prohibits the positive Δε LCs from reorienting upon application of the DC field. This enabled a 1000 nm red shift from 1200 to 2200 nm at 60 V. However, a strong AC field could be applied to overcome the alignment of the positive Δε LC to the polymer network, resulting in complete removal of the reflection notch. Blue shifting behavior was also attainable in this system by adjusting the curing conditions. The authors were able to fabricate a pixelated display with local control over the reflection color, as shown in Figure 4d–h.

### 2.3. Switching Response in PSCLCs

The effects of LC polymer composites can have dramatically different EO properties depending on their relatively low and high molecular weight, and the natural properties of each component [3,22]. In the case of PSCLCs, a bistable switching effect can be attained by using either positive Δε or negative Δε LC hosts, depending on the processing conditions. Bistable switching PSCLCs have two modes: normal mode [5,18,19,36,37] and reverse mode [5,20,21,36,37,38]. In normal mode, the PSCLC switches from a scattering state to a clear state. In reverse mode, the PSCLC shows a transition from a clear state to a scattering state. By altering the preparation conditions of the PSCLC sample, the two different switching modes can be achieved.

To prepare a PSCLC with positive Δε exhibiting normal mode switching, an electric field is applied during the photopolymerization to align the polymer network in the normal direction of the cell. Removal of the electric field after photopolymerization induces a focal conic scattering state of PSCLCs. This preparation method results in a clear state when an electric field is applied to the sample because the electric field orients the LCs into the homeotropic phase (clear state). In order to prepare a reverse switching mode sample, the PSCLC is prepared without applying an electric field during the photopolymerization. This causes the PSCLC to switch from a clear state to a stable scattering state when the electric field is applied.

Lee and coworkers presented a switchable PSCLC using negative Δε LCs [49]. Unlike positive Δε LCs, where the director rotates along the electric field direction, the negative Δε LCs in the planar geometry are not reoriented by an electric potential. The authors demonstrated the switching response of negative Δε PSCLCs upon the application of a DC field. The PSCLC samples were prepared with a low polymer concentration (<1.5 wt.%), leading to the generation of a weak polymer network. The polymer network was then detached from the alignment layer by applying a large electric field. The DC field shifts the polymer network towards the negative electrode, resulting in the switching behavior. Figure 5 shows the bistable switching of a PSCLC switched between clear and scatter states by varying the applied electric field to a maximum voltage of 200 V DC.

Figure 6 shows the response times of switchable PSCLCs with a thickness of 30 μm. Figure 6a shows the optical response of the sample to an applied DC and AC field. The initial scattering sample (Figure 6a(i)) became reflective when 210 V of DC voltage (forward direction) was applied (Figure 6a(ii)). The reflective state remains stable when the applied DC voltage is removed. A scattering state is then induced from the reflective state by the application of −45 V DC (reverse direction) (Figure 6a(iv)); the scattering state remains stable when the electric stimuli is removed. Interestingly, the reflective state could also be achieved by applying 100 V of an AC voltage at a frequency of 1 kHz (Figure 6(a(vi)). As shown in Figure 6a, the scattering and reflective states remain unchanged when the applied electric field is removed (Figure 6a(iii,v,vii)). In the scattering mode, the PSCLC has a transmittance less than 1%. In reflectance mode, the PSCLC exhibits transmittance values near 60% at a wavelength of 500 nm and a reflection notch in the infrared region. When the DC voltage is removed, the reflective mode persists and the scattering mode can be recovered upon the application of AC voltage. The average response time for switching from scattering mode to reflective mode was measured to be 200 ms, and switching from the reflective to scattering mode was measured to be 350 ms (Figure 6b).

A new switching mechanism that exploits the reflectance loss of PSCLCs in relatively thin cells using high DC fields was developed by Lee et. al. in 2020 [50]. This mechanism does not rely on the reorientation of positive Δε LCs or the electrophoretic movement of the polymer network when using negative Δε LCs. The PSCLCs were able to directly switch from a reflective state to a clear state with complete removal of the reflection notch upon application of a 45 V DC voltage. The PSCLC samples in this study had a much smaller thickness of 5 µm (Figure 7a). Interestingly, this method enabled control over the selective reflection position through moderate changes to the applied DC field. When a lesser voltage of 13 or 23 V was applied to the sample, the selective reflection notch reappeared. However, the reflection notch reappeared at a new position of 670 or 730 nm, respectively (Figure 7b,c). When the applied DC voltage is turned off, it shows a reversible behavior in which the reflection notch relaxes back to the initial notch position (Figure 7d). This study showed that the spectral position and reflectance of the sample could be controlled and turned off through the application of a DC voltage.

The ability to control the red shifting and switching behavior of the PSCLC sample was further refined using a small cell thickness. Figure 8 shows the red tuning and switching response of a thinner 3-µm thick PSCLC sample. The initial notch position of the sample was located at 500 nm; by applying a DC voltage of 23 V the reflection notch red shifts to 730 nm. At a higher applied voltage of 40 V, the reflection notch completely disappears, switching the sample from the reflective state to a clear state. After switching to the clear state, the reflection notch could be recovered by decreasing the applied voltage to 23 V. Interestingly, the position of the reflection was determined by the magnitude of the applied DC voltage (Figure 8a). The voltage-dependent tuning behavior not only led to a change in the reflected wavelength, but also an increase in the transmission of the sample, which is why switching to a clear state was attainable. The controllable tuning behavior at lower applied voltages and switching at higher voltages was also reversible (Figure 8b). Photographs of the PSCLCs reflection color at various DC voltages are shown in Figure 8c. The observed behavior of the thin PSCLCs was attributed to the deformation of the polymer network. As a DC voltage is applied to the cell, the polymer network shifts toward the electrode, which results in a decrease in the number of repeat units throughout the cell. This causes a red shift in the reflection notch as the PSCLCs pitch is elongated. A specific example is the 3-μm PSCLCs appearing transparent at an applied voltage of 40 V, where the limited number of repeated positions along the helical axis leads to loss of the reflection notch.

Lee et al. also determined that PSCLC samples with a smaller cell thickness had faster response times. Thin cell PSCLC samples exhibited a faster response time when switching from the scattering state to clear state compared to PSCLCs with increased thickness [47,50]. Figure 9 shows the response time for a 3- and 5-µm thick sample, respectively. Response times of 30 ms and 50 ms were observed for the 3- and 5-µm thick PSCLCs when switching from the scattering state to the clear state, respectively. Similarly, response times of ~150 ms and ~200 ms were observed for the 3- and 5-µm thick PSCLCs when switching from the clear state to the scattering state [50]. As the PSCLC thickness was increased to 15 µm, the response times for switching of the sample was determined to be 1.5 s and 3 s for both modes, respectively [47].

### 2.4. Other Effects on Dynamic Behavior of PSCLCs

A variety of factors influence the responsive behavior of PSCLCs and their specific applications. In addition to the components in the PSCLC mixture, mainly the chiral dopant concentration and mesogen type, the processing parameters of boundary conditions and purity influence the electro-optic response. This section will discuss the effects of cell thickness and ion density on the dynamic response of PSCLCs. In addition, the high reflectivity (>50%) of a PSCLC at oblique incidence is discussed.

#### 2.4.1. Cell Thickness

Modifying the cell thickness of PSCLCs has a dramatic effect on their reversible electro-optic response, including the reflection notch tuning range, in-band transmittance, and response time [50,55]. The pitch of the PSCLC samples prepared at various cell thicknesses was ~0.40 µm, as shown in Figure 10. Each PSCLC examined exhibited an electrically induced red shift of the reflection notch, but the tuning range was dependent on the thickness of the cell. The PSCLCs with larger cell thicknesses of 7.6~14.1 µm showed a red shift of ~400 nm at high DC voltages (55–90 V DC), as shown in Figure 10a–c. The transmittance of the reflection notch with a notch position of ~620 nm was <5% over the entire tuning range when the right-handed circularly polarized light (RH-CLP) was used as the probe beam. However, the reflection notch of PSCLCs with cell thicknesses ≤5 µm exhibited a narrower tuning range and higher transmittance, as shown in Figure 10d–f. At higher voltages, the PSCLCs with thinner cell thicknesses appear transparent due to the limited number of repeated positions along the helical axis, as the pitch of the PSCLC is deformed, resulting in a significant decrease in the reflection efficiency. This electric field increases the transparency of the cell because the number of repeat units for the extended pitch is insufficient to attain good reflection efficiency in the cell thickness. However, the thinner PSCLCs show a faster response of ~30 ms and ~50 ms rise times and ~150 ms and ~200 ms fall times for 3-µm and 5-µm thick samples, respectively, than those of thick samples, which were observed to be 1.5–3 s for a 15-µm thick sample.

#### 2.4.2. Ion Effect

Since the electro-optic response in PSCLCs is commonly attributed to an ion-facilitated electro-mechanical deformation [48,78], changes in the ion density have been found to influence the optical response. During photopolymerization of the polymer-stabilizing network, ions are formed from impurities in the LC monomers and the excess photoinitiator added to the mixture [78,79]. The residual ions that are formed act as impurities, enabling the deformation of the polymer network, subsequently altering the electro-optic response in PSCLCs. The effect of ion concentration on the dynamic response in PSCLCs is further discussed.

It has been documented that polymer networks can be formed from LC mixtures without the presence of photoinitiators [79,80]. The PSCLCs prepared without photoinitiators show lower ion concentrations and different electro-optic responses than PSCLCs prepared in the presence of photoinitiators. Lee and coworkers reported that the LC monomers C3M and C6M were first recrystallized in methanol and separated into two fractions; a purified fraction containing low impurities and low ion concentrations, and a residual fraction containing impurities and high ion concentration [79]. The PSCLCs prepared using the as-received LC monomers had initial ion densities of 7.6 × 10^13^ ions cm^−3^; these samples exhibited a bandwidth broadening response as the applied DC voltage was increased (Figure 11a). Interestingly, the PSCLCs prepared with recrystallized LC monomers had lower ion concentrations than the as-received LC monomers and exhibited a smaller bandwidth-broadening response (Figure 11b). While the PSCLCs prepared with the fully purified LC monomers showed a red tuning of the reflection notch instead of bandwidth broadening (Figure 11c). This behavior is attributed to the polymer network entrapping the ions after photopolymerization, the ions can then lead to deformations in the polymer network as the electric stimuli is applied and increased.

Further evidence that ion concentration directly impacts the electro-optic response of PSCLCs was discovered by Lee et al., where samples were exposed to UV light during the application of an electric stimuli. Here, PSCLCs were prepared using 1 wt.% of the photoinitiator Irgacure 369; the PSCLCs were then exposed to irradiation with a UV lamp during the application of a DC voltage. A red-orange reflection color was observed for the initial sample, which became a deep red color when the DC voltage was increased to 20 V. The color became a darker red, ultimately shifting into the infrared region when UV light was exposed during the application of the 20 V. Images of the PSCLCs at different conditions and the transmission spectra of the observed behavior are shown in Figure 12a and b. The observed behavior was attributed to the generation and activation of more ions upon exposure to UV light during the application of the electric field. The increased ions enabled a further redshift of the reflection notch and the resulting color change. Interestingly, UV exposure induces a reversible response.

The driving force for the electro-optic response in PSCLCs is thought to come from the movement of ionic impurities, which will be further discussed in Section 2.5. Upon the application of an electric field, the ionic impurities trapped within the polymer network migrate to the electrodes, inducing a polymer deformation. The deformation of the polymer-stabilizing network results in the tuning of the reflection band of the PSCLC. The reliance on ionic impurities to induce the reflection notch tuning could be a potential issue for utilizing PSCLCs in display technology, as it could induce image sticking. Image sticking is a phenomenon in which a previously displayed image is observed after the applied voltage has been removed and the display has been refreshed [81]. To alleviate the potential issue of image sticking, the choice of negative Δε LC and alignment layer would need to be optimized to reduce the affinity for the ions to adsorb to the alignment layer [81]. Another potential solution that has been shown to reduce image sticking is through doping of either the LC phase or alignment layer with nanoparticles such as γ-Fe2O3 [82] and TiO2 [83]. The doping method would need to be further explored in PSCLC systems, as the proposed mechanism for reducing image sticking relies on ion impurities being absorbed by the nanoparticles, which could reduce tuning ability.

#### 2.4.3. High Reflectivity of a PSCLC at Oblique Incidence

This type of structure results in unpolarized light transmitting one handedness through the CLC while reflecting the opposite handedness of light. In order to achieve total reflectance of unpolarized light, a CLC system must be designed to have CLC layers with different handedness, either creating physically distinct right-handed and left-handed regions within a single CLC sample [31,34,35,84,85,86,87,88] or stacking CLCs with different handedness [42,89,90]. The design and fabrication of these CLC systems often result in a myriad of problems limiting the effectiveness of the system. In order to minimize the problems arising from the difficult fabrication, Tondiglia et al. demonstrated an electrically controlled total reflection within a single-layer PSCLC by observing the reflection at oblique incidence angles [91]. The researchers observed a reflectance notch with full reflection of unpolarized light when the PSCLC device was tilted at an angle of 60° (Figure 13). The increased reflectance was attributed to an increased path length through the PSCLC; this also resulted in an effective refractive index change within the PSCLC layer, resulting in scattering losses. In addition to achieving full reflectance, the pitch of the PSCLC and bandwidth of the reflectance peak could be controlled by the application of DC fields. Figure 14 shows the bandwidth broadening and red tuning response of PSCLC samples at normal incidence and at an incidence angle of 60° as a function of the applied DC field. At a 60° incidence angle, both the bandwidth broadening and red tuning samples showed large electro-optic responses with the total reflection being much less than the 50% reflectance of the sample measured with a 0° incidence angle. The total reflectance behavior was achieved using a simple fabrication method through the alteration of the incidence angle, which enabled an electrically controlled total reflection in a single-layer PSCLC.

### 2.5. Potential Mechanism

The electrically induced optical response of PSCLCs is attributed to the deformation of the polymer network stabilizing in the CLC medium [39,41,46,50,79]. Electro-optic experiments reveal a strong correlation between the magnitude of the optical response and the concentration of ionic impurities in the LC host. In LCs or LC mixtures, ions exist in the range of 10^9^–10^14^ ions/cm^3^ and are residual impurities generated from initiators, catalysts, salts, moisture used during synthesis and purification processes [78]. Further studies have also determined that the UV curing process used to form the polymer stabilizing network can further increase the ion density due to degradation of the liquid crystals [92,93,94,95] and alignment layers [63]. Three types of ionic impurities can be present in LC mixtures: positive, negative and neutral ions. The positive ions in the mixture can be bonded to the ester group of the liquid crystal monomer and the polar group of the photoinitiator. During photopolymerization a fraction of the positively charged ions are trapped on or in the polymer network, represented by green charges, while some exist as free ions, represented as red positive and blue negative ions, as shown in Figure 15a. Application of a DC electric field causes charge screening, in which free ions present in the system migrate toward the oppositely charged electrodes. An increase in the DC field can then move the trapped ions in the polymer, leading to the deformation of the polymer network across the cell thickness direction, as can be seen in Figure 15b. If the polymer network is attached to the substrates while applying a DC field, the number of pitches must remain constant and the deformation of the polymer network induces a pitch variation across the cell; i.e., pitch compression and expansion near the negative electrode and the positive electrode, respectively. The degree of deformation of the polymer network is mainly influenced by two factors: the viscoelastic properties of the polymer network [46,47] and the type and concentration of ions trapped in the polymer network [48,79,80]. Such field-induced deformation and related pitch distribution depend on the nature of the liquid crystals, the polymers, the type and concentration of ionic impurities, and the chemical and physical properties of the crosslinked polymer network.

The mechanism of the responsive electro-optic behavior of PSCLCs being driven by the pitch modulation of the CLC medium through polymer deformation has been further experimentally supported. Pitch changes can be probed using confocal or multiphoton fluorescence polarization microscopy, which enables visualization of the cross-sectional image of PSCLCs [41,46]. Due to the LC molecules anchoring to the polymer network, the pitch of the CLC is deformed. The change in the pitch is directly observable using fluorescence confocal microscopy. Figure 16 and Figure 17 exhibit the fluorescence confocal microscopy images of PSCLCs, showing bandwidth broadening and red-tuning, respectively [41,46]. In both cases, applying a DC field results in non-uniform pitch distortion. In Figure 16, Nemati and colleagues were able to determine the helical pitch change of PSCLCs exhibiting bandwidth broadening [41] using an LC mixture consisting of 91.48% negative Δε LC HCCH, 2% chiral dopant R811, 6.5% monomer RM257, and 0.02% fluorescence dye BTBP, *N,N’*-bis(2,5-di-*tert*-butylphenyl)-3,4,9,10-perylenedicarboximide). The pitch of the resulting PSCLC was determined to be ~5 µm. Figure 16a shows the cross-sectional image of the florescence radiation intensity profile of the sample before the application of an electric field, and shows a layered structure having a uniform half pitch (P/2). When a DC field of 1.1 V µm^−1^ was applied, a non-uniform pitch change was observed. The CLC layers expanded near the positive (top) electrode and compressed near the negative (bottom) electrode, as shown in Figure 16b. When the polarity of the applied DC voltage was reversed, the CLC layers near the negative (top) electrode were compressed, while the CLC layers near the positive (bottom) electrode were expanded. However, it was observed that a large fraction of the CLCs in the bulk maintained their initial pitch. This observation indicates that, in the case of symmetrical broadening, most of the CLCs maintain their initial pitch, while a fraction of pitches are compressed near the negative electrode and expanded near the positive electrode.

Similar to responsive PSCLCs exhibiting bandwidth broadening, tunable PSCLCs are also driven by pitch modulation. The pitch of a PSCLC exhibiting red tuning is shown in Figure 17a; the period of the half-pitch starts to change near the top and bottom substrates by increasing the DC field [46]. Each line represents the half pitch of the CLC. At this low DC field, a pitch change similar to the bandwidth broadening sample is observed. As the electric field strength increases, most of the pitches in the bulk CLC expand, except for a small number of pitches that are compressed near the negative electrode. The corresponding half pitch variation in cell thickness is shown in Figure 17b. At a low DC field of 0.5 V µm^−1^, the pitch change occurs near the top and bottom of the substrate, with one side expanding and the other side contracting. At larger DC fields greater than 1.0 V µm^−1^, the pitch deviates from the original pitch value across the cell thickness and increases with the applied DC field. The sample highlighted in Figure 17 and similarly formulated PSCLCs show a red shift of the Bragg reflection peak with increasing the DC field, but no significant change in bandwidth [44,46].

## 3. Conclusions

In this review, the electrically induced optical response of PSCLCs fabricated with positive or negative Δε LCs was summarized. Polymer stabilization of the CLC phase enabled responsive behaviors not observed in bulk CLCs; specifically, the tuning of the selective reflection across the electromagnetic spectrum and bandwidth broadening. Large-magnitude dynamic bandwidth broadening of the reflection notch of PSCLCs was observed in PSCLCs with relatively low crosslink density of the polymer network, whereas a red shift or blue shift of the reflection band was observed in PSCLCs with higher crosslink densities. The bistable switching between reflective and scattering states of PSCLCs was attributed to an electromechanical polymer displacement mechanism. In thin cells, a reversible switching between reflective and transparent states was observed because the number of effective repeat units in the cholesteric pitch became too small to achieve measurable reflection efficiency. The potential mechanism for the dynamic responses of PSCLCs was attributed to the electric-field-induced polymer deformation in a CLC medium. Increasing the electric field controls the degree of polymer deformation, inducing a pitch variation across the cell thickness. The dynamic control of optical properties enables the PSCLCs to be used in several optical applications, including displays, optical filters, smart windows, and mirrors. The use of PSCLCs in these technologies requires addressing several potential challenges, such as high DC driving voltages, image sticking problems, and modest optical properties. In addition, the deformation of the polymer network to induce the dynamic response of the PSCLC results in a relatively slow response time in the order of seconds, lower transmittance, and increased haze. These practical issues of PSCLCs require further investigation and optimization in future studies.

## Data Availability

The data presented in this study are available on request from the corresponding author.

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
