# Peer review of "Recent Advances in Electro-Optic Response of Polymer-Stabilized Cholesteric Liquid Crystals"

_materials, 2023, doi:10.3390/ma16062248_

Round 1

Reviewer 1 Report

Reviewer’s Comments:

The manuscript “Electro-optic Response of Polymer Stabilized Cholesteric Liquid Crystals” is a very interesting work. In this work, Cholesteric liquid crystals (CLC) are a class of organic small molecules that self-assemble into helicoidal superstructures exhibiting circularly polarized reflection. The facile self-assembly and resulting optical properties makes CLCs a promising technology for an array of industrial applications including reflective displays, tunable mirror-less lasers, optical storage, tunable color filters, and smart windows. CLCs have the ability to stabilize the helicoidal structure via in-situ photopolymerization to generate a polymer stabilized CLC (PSCLCs). PSCLCs exhibit a dynamic optical response that can be induced by external stimuli including electric field, heat, and light. In this review, we discuss the electro-optic response and potential mechanism of PSCLCs reported over the past decade.The results are consistent with the data and figures presented in the manuscript. While I believe this topic is of great interest to our readers, I think it needs major revision before it is ready for publication. So, I recommend this manuscript for publication with major revisions.

1. In this manuscript, the authors did not explain the importance of the Liquid Crystals in the introduction part. The authors should explain the importance of Liquid Crystals.

2) Title: The title of the manuscript is not impressive. It should be modified or rewritten it.

3) Correct the following statement “The reconfigurable optical response of PSCLCs with positive dielectric anisotropy is also discussed. Finally, we discuss the potential mechanism for the dynamic electro-optic response in PSCLCs.”.

4) Keywords: There so many keywords and reduce them up to 5. So, modify the keywords.

5) Introduction part is not impressive. The references cited are very old. So, Improve it with some latest literature like 10.3389/fchem.2022.1023316, 10.1016/j.eurpolymj.2021.110783

6) The authors should explain the following statement with recent references, “However, polymer stabilization of these systems has largely corrected this issue”.

7) Add space between magnitude and unit. For example, in synthesis “21.96g” should be 21.96 g. Make the corrections throughout the manuscript regarding values and units.

8) The author should provide reason about this statement “During photopolymerization a fraction of the positively charged ions are trapped on or in the polymer network, represented by green charges, while some exist as free ions, represented as red positive and blue negative ions, as shown in Figure 15(a)”.

9) Comparison of the present results with other similar findings in the literature should be discussed in more detail. This is necessary in order to place this work together with other work in the field and to give more credibility to the present results.

10) Conclusion part is very long. Make it brief and improve by adding the results of your studies.

11) There are many grammatic mistakes. Improve the English grammar of the manuscript.

Reviewer 2 Report

This is a good reviewer paper about the electro-optical response of PSCLC. 

I have a few comments below:

1. Please give a more detailed summary in the conclusion of each section presented in this manuscript. The conclusion is too short in my view.

2. Some figures are not in high quality. Please try to download high quality figure from the original paper rather than screenshot. (Figure 8, 9,10, 14)

Author Response

We appreciate the reviewer’s insightful and helpful comments on our manuscript. The comments from the reviewer were very helpful in revising the manuscript. We have carefully considered and addressed all of the reviewer’s comments and suggestions. Corrections and our responses to reviewer’s suggestions and comments have been added to the manuscript. Please find our response in blue font and revisions in red font. In the revised manuscript, the changes are presented in red font.

Reviewer #2:

We appreciate the reviewer’s insightful and helpful comments.

1. Please give a more detailed summary in the conclusion of each section presented in this manuscript. The conclusion is too short in my view.

Response: A detailed summary has been added to the Conclusion section, such as “In this review, the electrically induced optical response of PSCLCs fabricated with positive or negative Δε LCs has been summarized. Polymer stabilization of the CLC phase enables responsive behavior not observed in bulk CLCs; specifically tuning of the selective reflection across the electromagnetic spectrum, bandwidth broadening, and bistable switching. Large-magnitude dynamic bandwidth broadening of the reflection notch of PSCLCs is observed in PSCLCs with relatively low crosslink density of the polymer network, whereas a red-shift or blue-shift of the reflection band is observed in PSCLCs with higher crosslink densities. The bistable switching between reflective and scattering states of PSCLCs is attributed to an electromechanical polymer displacement mechanism. In thin cells, a reversible switching between reflective and transparent states is observed because the number of effective repeat units in the cholesteric pitch becomes too small to achieve measurable reflection efficiency. The potential mechanism…

2. Some figures are not in high quality. Please try to download high quality figure from the original paper rather than screenshot. (Figure 8, 9, 10, 14)

Response: Higher quality Figures have been added.

Reviewer 3 Report

Dear Authors,

The review on recent results in PS CLC covers an interesting topic and may be of interest to expert society. However, there are several critical concerns to be addressed.

The review is poorly structured and should be improved. It would be much clearer to compare various designs or effects if one would stick to particular cell geometry – standing or lying helix, plane or in-plane electrodes, negative or positive LC dielectric anisotropy. Now all the designs are piled up and categorized according to electrooptic behaviour – bandwidth shift, bandwidth broadening, switching to scattering state, and later (suddenly) – lying helix, and standing helix. The last two modes are illustrated with self-citing only.

Please, take a look at a recent review of PS CLC: Guanjun Tan et al 2017. Review on polymer-stabilized short-pitch cholesteric liquid crystal displays // J. Phys. D: Appl. Phys. 50, 493001. DOI: 10.1088/1361-6463/aa916a. There you can see a clear structure: the first figure shows the geometries under study, then table 1 with possible ways to achieve a hard-to-obtain ULH geometry, below it is continued with specific measured characteristics, and finalized with table 2 of pros and cons of various geometries. On the other hand, you start from an elementary scheme 1 that is far out of interest to anyone who is aware of the LC phase existence. By the way, as a first reference, you could mention an earlier book on LC electrooptics [Blinov L.M. Electro-optical and Magneto-optical Properties of Liquid Crystals. New York: Wiley, 1983.]. In scheme 2, a selective reflection illustration is given and there are no other experiment geometry schemes in the paper. Thus, should the reader think that the standing helix is the only geometry under study?

After the introduction, there goes a bulk text citing pieces of the papers published by the Authors. All 17 figures are from the articles by the Authors. Should we believe that nobody else has worked on PS CLC in the last decade? Even if so, no attempt is made to compare the achieved results, show their pros and cons, figure out the direction of future findings. It is not a review, it’s just a summary of your recent papers. If you want to publish it, then clearly state it in abstract or title. E.g., change the title to “Our recent results on electrooptics of PS CLC”.

Also, there are a few misguiding sentences:

Line 17 – “CLCs have the ability to stabilize the helicoidal structure via in-situ photopolymerization to generate a polymer stabilized CLC.” The CLC itself does not have photopolymerization ability, the LC monomer is required to set the PS CLC.

Line 49 – light is exposed to a CLC layer. The “single PSCLC” does not sound like a common term.

Line 54 – P0 is a natural CLC pitch.

Lines 55, 67 – excessive parentheses.

I would go deeper into the text but later decided that your review requires a major revision.

My brief search with keywords returned many references related to your topic. If you wish to complete an essential review, I propose searching a literature once again and include the last decade results by other authors.

Round 2

Reviewer 3 Report

Dear Authors,

Here are a few items to be fixed through the text:

Line 36. The medium can be anisotropic, but the refractive indices are just numbers and ‘anisotropic refractive indices’ sounds like a slang. Please, rephrase. 

Line 37. Please, correct the sentence after deletion the Scheme 1.

Line 54. I don’t understand why you deleted Scheme 2. Now the paper does not contain ANY scheme of the experiment geometry.

Line 64. I think ‘the bandwidth of CLC selective reflection’ was meant here.

Line 67. ‘the long axis of the liquid crystal molecule’ was supposed to be here.

Line 93. Strongly -> strong

Line 219. Please, insert the LC cell gap value.

Line 239. Consider this work as a strong competitor to the PSLC technology: G. Nava, F. Ciciulla, F. Simoni, O. Iadlovska, O. D. Lavrentovich & L. Lucchetti (2021) Heliconical cholesteric liquid crystals as electrically tunable optical filters in notch and bandpass configurations, Liquid Crystals, 48:11, 1534-1543, DOI: 10.1080/02678292.2021.1884911

Line 316. Please, insert the LC cell gap value.

Line 342. Please, insert the LC cell gap value.

Line 366. Please, insert the LC cell gap value.

Lines 594-596. The same idea is written at Line 52. First two sentences of the chapter can be deleted without any loss.

Line 599. ‘Multiple regions’ – what kind of regions - in space? or spectral bands?

Line 620. Please, insert the LC cell gap value.

Line 626. Please, insert the LC cell gap value.

Line 727-728. Sounds too bold. The bistable switching is well-known in nematic liquid crystals. Feel free to google the ‘Nemoptic display technology’, for example. And it is not the only example.

As the observed effects are proposed for application, it would be useful to include into the discussion the ways proposed to alleviate the image sticking problem in case of application of large DC field.

Also, I would not be as optimistic as Authors in the last sentence of the Conclusions. I believe large driving voltage, image sticking problem and modest optical characteristics strongly impede the use of the proposed PSCLC technologies in such well-developed applications as displays, tunable optical filters and smart windows. Said problems should be stated clearly to the reader.
